# Design, Synthesis, and Tribological Behavior of an Eco-Friendly Methylbenzotriazole-Amide Derivative

**DOI:** 10.3390/ijms26031112

**Published:** 2025-01-27

**Authors:** Fan Yang, Zenghui Li, Hongmei Yang, Yanan Zhao, Xiuli Sun, Yong Tang

**Affiliations:** 1School of Materials and Chemistry, University of Shanghai for Science and Technology, Shanghai 200093, China; yangfan@sioc.ac.cn (F.Y.); lizenghui@mail.sioc.ac.cn (Z.L.); 2State Key Laboratory of Organometallic Chemistry, Shanghai Institute of Organic Chemistry, Chinese Academy of Sciences, Shanghai 200032, China; yananzhao99@sioc.ac.cn (Y.Z.); xlsun@sioc.ac.cn (X.S.); tangy@mail.sioc.ac.cn (Y.T.)

**Keywords:** multifunctional additive, methylbenzotriazole, amide derivative, synthetic hydrocarbon, tribological behavior

## Abstract

Recently, researchers have been committed to boosting the environmental friendliness and functional performance of multifunctional additives. In this study, an eco-friendly methylbenzotriazole-amide derivative (MeBz-2-C18) was designed and synthesized, with ethylamine serving as the linkage between methylbenzotriazole and the oleoyl chain. The structure of MeBz-2-C18 was characterized by nuclear magnetic resonance (NMR), high-resolution mass spectrometry (HR-MS), Fourier-transform infrared spectroscopy (FT-IR), and thermogravimetric analysis (TGA). Subsequently, the storage stability and tribological behavior of MeBz-2-C18 and the commercial benzotriazole oleamide salt (T406) were comparatively evaluated. The covalently-bonded MeBz-2-C18 exhibits superior thermal stability, along with boosted storage stability and tribological performance in the synthetic base oil. Specifically, 0.5 wt.% addition of MeBz-2-C18 and T406 can reduce the average wear scar diameter (ave. WSD) by 21.6% and 13.9%, respectively. To further explore the micro-mechanism, the electrostatic potential (ESP) and worn surfaces were analyzed with scanning electron microscope-energy dispersive spectrometer (SEM–EDS), X-ray photoelectron spectroscopy (XPS), and density functional theory (DFT) calculations. The results show that MeBz-2-C18 possesses stronger adsorption on the metal surface, and its amide bond preferentially breaks during friction. This reduces the interfacial shear force and promotes the film formation of iron oxides, thus resulting in superior tribological performance.

## 1. Introduction

As mechanical equipment upgrades, modern lubricants usually contain additives to meet the demanding lubrication requirements under higher temperatures, loads, or speeds [1]. Lubricant manufacturers are currently interested in promising new multifunctional lubricants with good functional properties [2]. Due to their dense chemical structures, good thermal stability, and ease of adsorption on metal surfaces, nitrogen-containing heterocyclic compounds can improve the extreme pressure, anti-wear, friction-reducing, and anti-corrosion properties of lubricants, which are applied in a variety of multifunctional additives [3,4,5,6]. Moreover, nitrogen can form efficient protective films during friction and be absorbed by microorganisms as a primary nutrient after use, contributing to their biodegradability [7,8,9]. Among them, benzotriazole and its derivatives are commonly used additives that not only exhibit excellent tribological properties but also possess anti-oxidation, anti-corrosion [10], rust prevention [11], and dispersion capabilities [12].

In 2002, Li et al. [13] replaced the metal ions in dialkyldithiophosphate with benzotriazole groups to synthesize sulfur- and phosphorus-containing benzotriazole derivative (BDDP), which exhibited good extreme pressure and friction-reducing properties in rapeseed oil. The worn surface analysis revealed that BDDP underwent tribochemical reactions with the metal surface during friction, in which the S and P elements transformed inorganic metal salts such as FeSO_4_, FePO_4_, etc., while the N element was mainly adsorbed on the metal surface in the form of organic amines. These two components together formed the boundary lubricating film, improving the tribological behavior of the base oil. With the deepening of the concept of sustainable development and the understanding of green chemistry, researchers have started to focus on non-metallic and environmentally friendly additives. Therefore, Xiong et al. [14] prepared two green imidazoline derivatives, benzotriazole-containing imidazoline (BML) and caprylic acid-containing imidazoline (CML), and conducted a comparative study on their performance as multifunctional additives. The results showed that both BML and CML exhibited good extreme pressure and anti-wear properties, while the anti-wear performance of BML is better than CML for the double heterocyclic structure, and the friction-reducing property of CML is better than BML for the long chain fatty acid, indicating that the molecular structure is the key to regulating the performance of lubricating additives.

Although benzotriazole derivatives display good tribological performance as lubricating additives, the poor solubility in hydrocarbons limits their applications [15]. How to boost the oil solubility of benzotriazole derivatives has become a research hotspot. Li et al. [16] prepared a methylbenzotriazole fatty amine salt from methylbenzotriazole and fatty amine. When 0.8 wt.% of commercially available benzotriazole fatty amine salt was added to Group III mineral oil (150 N), the oil sample began to change from clear to cloudy. However, the prepared methylbenzotriazole fatty amine salt remained clear and transparent, indicating it had better oil solubility in the base oil, which exhibited better anti-corrosion performance. Additionally, in our previous research [17,18], we found that polar amide and ester bonds are more likely to break during friction. This reduces the interfacial shear force and enables easy reactions with metal surfaces to form iron oxide films, thus showing better anti-wear and friction-reducing performance.

In this study, aiming to improve the compatibility and functional performance of benzotriazole in hydrocarbons, a novel methylbenzotriazole-amide derivative (MeBz-2-C18) was innovatively designed and synthesized, using ethylamine as the linkage. The structure of MeBz-2-C18 was characterized by nuclear magnetic resonance (NMR), high-resolution mass spectrometry (HR-MS), Fourier-transform infrared spectroscopy (FT-IR), and thermogravimetric analysis (TGA). Subsequently, the storage stability and tribological behavior of MeBz-2-C18 and the commercially available T406 with similar composition were comparatively evaluated in the low-viscosity synthetic base oil. The worn surfaces were analyzed via scanning electron microscope-energy dispersive spectrometer (SEM–EDS) and X-ray photoelectron spectroscopy (XPS). These experimental analyses were integrated with density functional theory (DFT) calculations to elucidate the micro-lubrication mechanism of MeBz-2-C18.

## 2. Results

### 2.1. Synthesis

5-methylbenzotriazole-2-ethylamine (MeBz-2-Etn) was first synthesized through a substitution reaction between 2-chloroethylamine and 5-methylbenzotriazole, which was reacted directly with oleoyl chloride to prepare the amide compound MeBz-2-C18. The synthesis pathway is illustrated in Figure 1.

NMR, HR-MS, and FT-IR were used to determine the structure of MeBz-2-C18, and the details are presented in Section 3.2. The ^1^H NMR and FT-IR spectra of MeBz-2-Etn and MeBz-2-C18 are compared in Figure 1. In the ^1^H NMR spectrum of MeBz-2-C18, the hydrogen signals of oleoyl alkyl and –CH=CH– appeared at 2.13, 2.01, 1.55, 1.26, 0.88 ppm and 5.34 ppm, respectively (in Figure 1a). Meanwhile, the –CH_2_ directly connected to –NH underwent a significant downfield shift (from 3.36 ppm to 3.93 ppm) due to the introduction of C=O. As can be observed from FT-IR spectra in Figure 1b, compared with MeBz-2-Etn, there are no N–H stretching vibrations of primary amine at 3372.9 and 3295.8 cm^−1^ in MeBz-2-C18. However, the C=O(NH) characteristic stretching vibration of secondary amide appears at 1643.5 cm^−1^, and the N(CO)–H and C(O)–N(H) stretching vibration peaks of secondary amide appear at 3309.7, 3071.7 and 1265.1 cm^−1^, respectively. Moreover, the peak located at 721.8 cm^−1^ can be attributed to the in-plane bending vibration of the long alkyl chain –(CH_2_)_n_–. Collectively, these spectral characteristics serve as evidence for the successful grafting of the oleoyl group in MeBz-2-C18.

### 2.2. Stability Characterization

#### 2.2.1. Thermal Stability

Thermal stability serves as a crucial determinant in the lifespan of additives, which is an important guarantee for additives. In this regard, a comparative investigation was carried out on the thermal stabilities of MeBz-2-Etn, MeBz-2-C18, and the commercially available T406. According to the analysis of the thermogravimetric curves in Figure 2 and Table 1, the initial decomposition temperatures of T406, MeBz-2-Etn, and MeBz-2-C18 were measured to be 76.8 °C, 19.7 °C and 185.5 °C, while the maximum decomposition temperatures were determined as 221.1 °C, 217.4 °C and 374.4 °C, demonstrating that the thermal stabilities can be ranked as MeBz-2-C18 > T406 > MeBz-2-Etn in descending order. Furthermore, in comparison with T406, the residual masses of MeBz-2-C18 at 200 °C and 300 °C were found to be 99.9%, 79.9% and 89.3%, 11.0%, respectively, which indicates that the incorporation of ethylamine contributes to the thermal stability enhancement of additives.

#### 2.2.2. Storage Stability in Base Oil

The storage stability of lubricants is a decisive factor influencing the compatibility between additives and base oils, which also serves as a fundamental prerequisite for practical applications. Therefore, the storage stability of oil samples at room temperature was investigated to study the compatibility of MeBz-2-C18 in the synthetic base oil, which was also compared with the commercially available T406 of similar composition. As shown in Figure 3a, the oil samples with 0.25–1 wt.% additions of MeBz-2-C18 remained clear and transparent after being stored for 30 days, which is similar to the base oil (0 wt.%). In contrast, the oil sample with 0.5 wt.% T406 became cloudy on the seventh day (Figure 3b), indicating that the compatibility between MeBz-2-C18 and the synthetic base oil is better than that of T406. This may be due to the fact that the bonding modes of T406 and MeBz-2-C18 are ionic bonds and covalent bonds, respectively. The relatively weaker polarity of the covalent bond leads to better compatibility with non-polar base oils.

### 2.3. Tribological Behavior

#### 2.3.1. Different MeBz-2-C18 Additions

As for lubricating oils, their performance often varies with different additive additions, and the optimal addition can endow oils with the best overall performance. Figure 4 shows the tribological performance of oil samples with different MeBz-2-C18 additions. It can be seen from Figure 4a that there are significant differences in the friction curves between the base oil (0 wt.%) and oil samples with MeBz-2-C18. The friction running-in period of the base oil is long (~300 s), and the coefficient of friction (COF) in the relative-stable period fluctuates significantly. In contrast, the oil sample with 1 wt.% MeBz-2-C18 has a shorter running-in period (~200 s), and the COF is more stable and much smaller. A further comparison in Figure 4b shows that the oil with 0.5 wt.% MeBz-2-C18 has the smallest ave. WSD, which is decreased by 21.6% compared to that of the base oil, and its ave. COF reduces by 3.2%. The oil with 1 wt.% MeBz-2-C18 reveals the lowest ave. COF, which is reduced by 8.4% compared to that of the base oil, and the ave. WSD is decreased by 13.9%. Overall, the oil with 0.5 wt.% MeBz-2-C18 exhibits the best tribological performance compared to the base oil.

#### 2.3.2. Comparison with the Commercial Additive

The tribological performance of MeBz-2-C18 was compared with T406, using 0.5 wt.% as the optimal addition. As shown in Figure 5a, the COF of the base oil suddenly increased and significant fluctuations occurred after 1200 s during the friction. However, the COF of the oils with 0.5 wt.% T406 or MeBz-2-C18 showed smaller variations, indicating that the addition of T406 and MeBz-2-C18 could stabilize the friction. Compared with the base oil (Figure 5b), the ave. COF of the oil with 0.5 wt.% MeBz-2-C18 decreased by 3.2%, while that of the oil with 0.5 wt.% T406 increased by 4.5%. The ave. WSD of the oil with 0.5 wt.% addition of MeBz-2-C18 and T406 was decreased by 21.6% and 13.9%, respectively. The results indicate that MeBz-2-C18 exhibits better friction-reducing and anti-wear performance than that of T406.

### 2.4. Lubrication Mechanism

#### 2.4.1. Worn Surface Analysis

In order to investigate the micro-lubrication mechanism of the synthesized MeBz-2-C18 in the synthetic base oil, SEM–EDS and XPS were employed to analyze the composition of tribofilms on the worn surface and non-worn surface lubricated with the oil containing 0.5 wt.% MeBz-2-C18 (denoted as MeBz-2-C18_non-wear and MeBz-2-C18_wear, respectively). These were compared with those lubricated with the base oil (Base oil_wear).

The analysis results of morphologies and elemental compositions are presented in Table 2. From the SEM images, the metal surface lubricated by the base oil shows more severe wear than that with the oil containing 0.5 wt.% MeBz-2-C18, indicating MeBz-2-C18 enhances the synthetic base oil’s anti-wear performance. By EDS analysis, the Fe, C, and O contents of the base oil’s non-wear and wear surfaces are 69.95%, 22.54%, 7.51%, 45.97%, 25.57%, and 28.46%, respectively. Meanwhile, the Fe contents of the non-wear and wear surfaces were lubricated with oil containing 0.5 wt.% MeBz-2-C18 are 63.86% and 35.62%, lower than those of the base oil. The C and O contents of the 0.5 wt.% MeBz-2-C18 oil’s non-wear and wear surfaces are 27.57%, 34.24% and 8.58%, 30.14%, higher than those of the base oil, suggesting MeBz-2-C18 participates in the formation of friction films.

XPS was utilized to compare the bonding states of elements before and after tribological tests, which can offer valuable information regarding the chemical changes taking place during friction [19]. Figure 6 and Table 3 display the deconvolution analysis of C, O, Fe, and N for the worn surfaces lubricated with the base oil and oil containing 0.5 wt.% MeBz-2-C18. In the C1s spectra, there are four main peaks for MeBz-2-C18_wear, namely metal carbides (283.4 eV, ~2.9%), C–C/C=C (284.8 eV, ~72.9%), C–O/C–N (286.0 eV, ~20.4%) and C=O (288.5 eV, ~3.6%) [20,21]. In contrast, the Base oil_wear spectrum only exhibits C–C/C=C (284.8 eV, ~86.9%) and C–O/C–N (286.0 eV, ~13.1%). In the O1s spectra, peaks at 529.6 eV, 531.7/531.9 eV, and 533.1 eV correspond to Fe–O, C=O, and C–O bonds, respectively. Notably, the Fe–O and C–O content in MeBz-2-C18_wear increases compared with Base oil_wear, i.e., 30.0% vs. 13.0% and 6.7% vs. 0%, respectively. While the C=O content in MeBz-2-C18_wear decreases compared with Base oil_wear (63.3% vs. 86.9%). The results indicate that the amide bonds were broken during friction, which further led to the formation of iron oxides. The peaks at 709.6 eV, 723.5 eV, 713.5 eV, and 711.0 eV in the Fe2p spectra correspond to Fe (2p3/2), Fe^2^^+^ (2p1/2), Fe^2^^+^ (2p3/2) and Fe^3^^+^ (2p3/2), respectively. This suggests that local high temperatures and high loads caused chemical reactions between iron in steel balls and oxygen in the air during friction [22,23]. Combined with the O1s spectra, it is evident that an iron oxide film was formed during friction, which might be composed of Fe_2_O_3_, FeOOH, FeO, and Fe_3_O_4_ [24,25]. Additionally, the N1s peaks for MeBz-2-C18_wear appear at 399.4 eV and 401.7 eV, corresponding to N–O and C–N bonds, which indicates that some amides were converted into nitrogen oxides [25,26]. The results demonstrate that the friction film formed by MeBz-2-C18 consists of organic oxides and iron oxides, which can enhance the anti-wear and friction-reducing properties.

#### 2.4.2. DFT Calculations

To explore the influence of structure on the tribological performance of lubricants, DFT calculations were carried out using Gaussian16 software to analyze the electrostatic potential (ESP). The B3LYP hybrid functional was employed for the geometric optimization of MeBz-2-C18 and T406. The ESP of C, H, O, and N atoms was analyzed with the 6–31G(d) basis set. The optimized structures were characterized by harmonic vibrational frequencies, being identified as minima (Nimag = 0) or transition states (Nimag = 1). The electronic energy spectrum of the compounds was calculated by means of Multiwfn, with the molecular surface as an isosurface of electron density r = 0.001 a.u.

As shown in Figure 7, the minimum and maximum ESP values for T406 and MeBz-2-C18 are −0.0475, 0.0565, and −0.0665, 0.0731, respectively. According to [27], the smaller the minimum value of ESP, the stronger the adsorption capacity on the metal surface. This indicates that MeBz-2-C18 has a stronger adsorption ability on the metal surface compared with T406. Combined with the XPS analysis, it can be seen that the amide bond in MeBz-2-C18 breaks during friction. This is conducive to its reaction with the metal surface and the film formation of metal oxides, thus playing an anti-wear role. Moreover, the cleavage of the amide bond decreases the interfacial shear force and boosts the friction-reducing performance. Consequently, MeBz-2-C18 demonstrates superior tribological performance compared to the commercial benzotriazole oleamide salt.

Based on the worn surface analysis and DFT calculations, the interaction of MeBz-2-C18 with the friction counterpart is speculated as depicted in Figure 8. That is, the nitrogen-containing heterocycle and polar amide bond in MeBz-2-C18 interact with the metal surface, and its long alkyl chains stretch out in the base oil to form a protective film. The amide bond in MeBz-2-C18 is prone to break during friction, reducing the interfacial shear force and facilitating the film formation of iron oxides.

## 3. Materials and Methods

### 3.1. Materials

5-Methylbenzotriazole (98%) was obtained from J&K Scientific (Beijing, China). Chloroethylamine hydrochloride (98%) and 4-dimethylaminopyridine (DMAP, 99%) were sourced from Beijing InnoChem Science & Technology Co., Ltd. (Beijing, China). Tetrabutylammonium bisulfate (TBAS, 99%) and oleoyl chloride (89%) were purchased from Energy Chemical (Shanghai, China). Sodium hydroxide (NaOH) and triethylamine (Et_3_N, 99.9%) were acquired from Shanghai Laboratory Reagent Co., Ltd. (Shanghai, China). Dichloromethane (DCM, 99.9%) was supplied by Shanghai Titan Technology Co., Ltd. (Shanghai, China), and tetrahydrofuran (THF, 99.9%) was obtained from Changshu Hongsheng Fine Chemical Co., Ltd. (Changshu, China).

Durasyn^®^164 (PAO4, INEOS, London, UK) and Priolube 3970 (3970, CRODA, Snaith, UK) were separately purchased from Shanghai Qicheng Industrial Co., Ltd. (Shanghai, China) and Hersbit Chemical Co., Ltd. (Shanghai, China), which were applied as the base oil for the tribological evaluation of MeBz-2-C18 and T406 (Suzhou Xingchangrun Chemical Co., LTD), as a commercial additive.

### 3.2. Synthesis of MeBz-2-Etn and MeBz-2-C18

#### 3.2.1. Synthesis of MeBz-2-Etn

5-Methylbenzotriazole (10.00 g, 75.18 mmol), NaOH (9.02 g, 225.54 mmol), and THF (150 mL) were mixed in a flask, which reacted at room temperature for 30 min. Subsequently, 2-chloroethylamine hydrochloride (13.08 g, 117.77 mmol) and TBAS (1.06 g, 3.13 mmol) were added, and then the mixture was heated to reflux at 85 °C for 12 h. The reaction was monitored by thin-layer chromatography (TLC) until 5-methylbenzotriazole was completely converted. The reaction mixture was filtered, and the filtrate was evaporated to dryness. The residue was dissolved in DCM and washed with a saturated NaCl solution. The organic phase was extracted with DCM, and dried, and the crude product was purified by column chromatography (eluent: V_DCM_/V_MeOH_ = 7:1), obtaining MeBz-2-Etn as a colorless oily liquid (4.37 g, yield 33%).

^1^H NMR (400 MHz, Tetrachloroethane-d2) δ 7.75 (d, J = 8.7 Hz, 1H), 7.61 (s, 1H), 7.23 (d, J = 10.2 Hz, 1H), 4.76–4.69 (m, 2H), 3.36–3.31 (m, 2H), 2.48 (s, 3H), 1.75 (s, 2H) (Appendix A). ^13^C NMR (101 MHz, Tetrachloroethane-*d*_2_) δ 144.90, 143.02, 136.75, 129.45, 117.52, 116.38, 74.34, 74.06, 73.79, 59.45, 42.05, 22.25 (Appendix A). HR-MS (ESI) calcd. for C_9_H_13_N_4_ [M+H]^+^: 117.11328, found: 117.11328 (Appendix A). FT-IR (ATR): ν = 3369.7, 3034.8, 2945.6, 2866.9, 1904.8, 1563.3, 1448.8, 732.0, 598.1 cm^−1^ (Figure 1b and Appendix A).

#### 3.2.2. Synthesis of MeBz-2-C18

Within a flask under an argon atmosphere, MeBz-2-Etn (4.5 g, 25.57 mmol), DMAP (0.3 g, 2.56 mmol), and Et_3_N (6.47 g, 63.93 mmol) were mixed with anhydrous DCM (90 mL). Subsequently, oleoyl chloride (11.54 g, 38.36 mmol) was added slowly at 0 °C. Thereafter, the reaction mixture was allowed to react at 25 °C for a duration of 12 h. TLC monitoring was continuously carried out to confirm the complete conversion of MeBz-2-Etn. Once the conversion was verified, the reaction was quenched by the addition of water. The resulting mixture was successively washed with saturated NaHCO_3_ and NaCl solutions. The organic phase was then repeatedly extracted with DCM and dried over anhydrous Na_2_SO_4_. Finally, the crude product was purified by column chromatography (eluent: V_PE_/V_EA_ = 1:1), affording MeBz-2-C18 as a pale yellow solid (10.79 g, yield 96%).

^1^H NMR (400 MHz, Tetrachloroethane-*d*_2_) δ 7.77 (d, *J* = 8.7 Hz, 1H), 7.63 (s, 1H), 7.27 (d, *J* = 8.8 Hz, 1H), 6.06–6.02 (m, 1H), 5.34 (s, 2H), 4.85–4.79 (m, 2H), 3.93–3.89 (q, *J* = 5.7 Hz, 2H), 2.51 (s, 3H), 2.13 (t, *J* = 7.6 Hz, 2H), 2.01 (s, 4H), 1.55 (m, 2H), 1.26 (s, 20H), 0.88 (t, *J* = 6.7 Hz, 3H) (Appendix A). ^13^C NMR (101 MHz, C_2_D_2_Cl_4_) δ 173.53, 144.93, 143.06, 136.96, 130.20, 129.98, 129.64, 117.55, 116.40, 74.33, 74.05, 73.78, 55.70, 38.99, 36.66, 32.05, 29.92, 29.87, 29.68, 29.47, 29.40, 29.29, 29.26, 27.36, 27.33, 25.66, 22.88, 22.27, 14.40 (Appendix A). HR-MS (ESI) calcd. for C_27_H_44_N_4_ONa [M+Na]^+^: 463.34073, found: 463.34073 (Appendix A). FT-IR (ATR): ν = 3307.4, 3071.4, 2922.3, 2851.4, 1638.9, 1465.9, 721.8, 597.9 cm^−1^ (Figure 1b).

### 3.3. Characterizations

NMR characterization, including ^1^H NMR and ^13^C NMR, was conducted on a 400-MR (Varian, Palo Alto, CA, USA) using tetrachloroethane-d_2_ as the solution. HR-MS was carried out on JMS-T100LPAccuTOF LC-plus 4G (Nippon Electronics Corporation, Tokyo, Japan) using electrospray ionization. Nicolet iN10MX (Thermo Fisher, Waltham, MA, USA) was applied to record FT-IR spectroscopy by scanning from 400 to 4000 cm^−1^. TGA was performed on Q500 (TA, Milford, MA, USA) under a N_2_ atmosphere with a flow rate of 60 mL/min and a heating rate of 10 °C/min from 25 to 600 °C. The morphology and elemental composition of metal surfaces are analyzed by SEM–EDS using QUANTAX (Bruker, San Jose, CA, USA). The chemical state of specific elements and potential tribochemical films formed on the worn surface were analyzed using XPS (Thermo Scientific K-Alpha, Waltham, MA, USA) using an Al-Kα radiation source, and the obtained data was further analyzed with the Advantage 5.9931 software.

### 3.4. Preparation of Oil Samples

The low-viscosity synthetic hydrocarbon PAO4 (90 wt.%) and saturated polyol ester 3970 (10 wt.%) were heated and stirred at 60 °C for 2 h to obtain the blended base oil. 0.25–1.0 wt.% of self-synthesized MeBz-2-C18 or purchased T406 was added to the blended base oil, and the mixture was stirred at 60 °C for 2 h to prepare the oil samples containing additives.

### 3.5. Tribological Testing

The tribological behavior of MeBz-2-C18 in base oil was evaluated using a Tenkey MS10A four-ball testing machine (Xiamen Tenkey Automation Co., Ltd., Xiamen, China), which was compared with the commercially available T406. All balls used were made of GCr15-bearing steel with a diameter of 12.7 mm. Following the NB/SH/T 0189-2017 [18] standard, tests were conducted at 75 °C with a steel ball rotational speed of 1200 rpm and a load of 392 N for 1 h. Each test was performed at least three times to ensure the reproducibility of the ave. COF and ave. WSD.

## 4. Conclusions

In this study, ethylamine was innovatively utilized as the linker between methylbenzotriazole and the oleoyl chain to synthesize an eco-friendly MeBz-2-C18. The structure of MeBz-2-C18 was characterized using NMR, HR-MS, FT-IR, and TGA. The storage stability and tribological performance of MeBz-2-C18 in synthetic base oil were explored and compared with that of the commercial additive T406 of similar composition. Additionally, the micro-lubrication mechanism of MeBz-2-C18 was revealed. The main conclusions are as follows:(1)The covalently-bonded amide bond endows MeBz-2-C18 with superior thermal stability and better storage stability in synthetic base oil compared with T406. The residual mass of MeBz-2-C18 at 300 °C is 89.3%, while that of T406 is merely 11.0%.(2)The optimal addition of MeBz-2-C18 in the selected synthetic base oil is 0.5 wt.%, which reduces the ave. WSD and ave. COF by 21.6% and 3.2%, which is better than that of the commercial additive T406.(3)Worn surface analysis and DFT calculations suggest that MeBz-2-C18 has stronger adsorption on the metal surface than that of T406. The amide bond in MeBz-2-C18 breaks preferentially during friction, reducing the interfacial shear force and facilitating the film formation of iron oxides.

## Data Availability

Data are contained within the article and Appendix A.

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
