# Peer review of "Design, Synthesis, and Tribological Behavior of an Eco-Friendly Methylbenzotriazole-Amide Derivative"

_ijms, 2025, doi:10.3390/ijms26031112_

Round 1
Reviewer 1 Report
Comments and Suggestions for Authors
1. The abstract seems to be lacking crucial information, including an outline of the results that will be presented in the study, the methodology used to achieve those results, a clear description of the objectives, and a comprehensive discussion of the approach taken in each stage of the research.
2. Literature survey in the introduction is relatively poor and the references are relatively old. The introduction has to be improved by addressing the more recent works.
3. The research gap is not properly identified and explained in the introduction section.
4. The main objectives and innovation of the paper must be written in a clearer and more concise way at the end of the introduction section.
5. The "Materials and Methods" section should be located prior to the "Results" section.
6. The description of the synthesis of MeBz-2-Etn and MeBz-2-C18 is brief and lacks detail. It would be helpful to provide more information about the reaction conditions, such as temperature, reaction time, and yield.
7. Figures 1 and 2 lack explanations of their contributions to the findings. A brief description of each figure would improve understanding.
8. The manuscript mentions several figures and tables, such as Figure S3 and Table S3, that are only found in the supplementary file and not included in the main text. This discrepancy may confuse readers and should be addressed for clarity.
9. The supplementary file should be integrated into the manuscript for better coherence and accessibility.
10. The importance of additive-base oil compatibility for storage stability is not sufficiently explained.
11. Figure 3 and Table S1 are referenced without explanation, and the implications of the compatibility results are not discussed.
12. Provide background information on the commercial additive T406 and its relevance to the study. Explain the importance of comparing MeBz-2-C18 with T406.
13. Present a more explicit comparison between MeBz-2-C18 and T406, highlighting the differences in COF and WSD percentages.
14. Offer a clear explanation of Figure 5, its content, and how it supports the claims made in the text.
15. Clearly explain the chemical changes observed in XPS analysis, including the increased C=O content and the appearance of N-O and C-N bonds in MeBz-2-C18 samples.
16. The comparison between MeBz-2-C18 and T406 is mentioned, but the significance of the differences in ESP values is not thoroughly discussed.
17. Section 3.2: The text is presented as a continuous block of text without clear headings, subheadings, or bullet points, making it difficult to read and comprehend. Improving the formatting and structure would make the text more accessible and user-friendly.
18. The conclusions section requires significant improvements and should be effectively rewritten after addressing the aforementioned comments.
19. The paper shall be logically rearranged and rephrased to help understanding the main point of this work, as well as highlighting novelty. The current manuscript is more like a technical report rather than a research paper. The research contents are listed linearly, without highlighting the focus and innovation.
Author Response
- The abstract seems to be lacking crucial information, including an outline of the results that will be presented in the study, the methodology used to achieve those results, a clear description of the objectives, and a comprehensive discussion of the approach taken in each stage of the research.
Response: We are grateful for your invaluable suggestions. In the revised version, we have thoroughly reorganized the abstract and added essential details, including a concise summary of the research findings, the methods utilized to achieve those results, a clear statement of the research objectives, and a comprehensive exploration of the approaches adopted in each stage of the research. Please see the abstract in the revised manuscript.
- Literature survey in the introduction is relatively poor and the references are relatively old. The introduction has to be improved by addressing the more recent works.
Response: Thanks for your proposal. We have carefully surveyed the research works in the relevant field, and rewritten the introduction. The following are the newly-addressed recent references. For details, please see the revised introduction in Section 1.
[1] Danilov AM, Bartko RV, Antonov SA. Current Advances in the Application and Development of Lubricating Oil Additives. Pet. Chem. 2020, 61, 35-42.
[2] Naletova AV, Davydov DV, Bakunin VN. Derivatives of 2,5-Dimercapto-1,3,4-Thiadiazole as Multifunctional Lubricant Additives. Chem. Technol. Fuels Oils 2021, 57, 783-791.
[3] Xiong, L.; He, Z.; Han, S.; Tang, J.; Wu, Y.; Zeng, X. Tribological Properties Study of N-Containing Heterocyclic Imidazoline Derivatives as Lubricant Additives in Water-Glycol. Tribol. Int. 2016, 104, 98-108.
[4] Ding, J.; Fang, J.; Chen, B.; Zhang, N.; Fan, X.; Zheng, Z. Improved Biodegradability and Tribological Performances of Mineral Lubricating Oil by Two Synthetic Nitrogenous Heterocyclic Additives. Ind. Lubr. Tribol. 2019, 71 (4), 578-585.
[5] Chen, Q.; Zhang, R.; He, Z.; Xiong, L. Tribological Performance of N-Containing Heterocyclic Triazine Derivative as Lubricant Additive in Ethylene Glycol. Surf. Interface Anal. 2021, 53 (12), 1027-1034.
[6] Xiong, S.; Wu, H.; Liu, Z. N-Containing Heterocyclic Benzotriazole Derivatives as New Corrosion Inhibitor for Mild Steel Contained in Emulsion. Anti-Corros. Methods Mater. 2022, 69 (2), 183-193.
[7] Wang, X.; Xu, P. Microbial Degradation of Nitrogen Heterocycles. Int. Biodeterior. Biodegrad. 2019, 142, 170-171.
[8] Lu, Q.; Zhang, Y.; Li, N.; Tang, Y.; Zhang, C.; Wang, W.; Zhou, J.; Chen, F.; Rittmann, B. E. Using Ultrasonic Treated Sludge to Accelerate Pyridine and p-Nitrophenol Biodegradation. Int. Biodeterior. Biodegrad. 2020, 153, 105051.
[9] Piccioni, M.; Varesano, A.; Tummino, M. L. Behavior of Polypyrrole-Coated Cotton Fabric Undergoing Biodegradation in Compost-Enriched Soil. Environ. Res. Commun. 2024, 6 (6), 065001.
[10] Adavodi, R.; Dini, G. Benzotriazoium Bis(2-Ethylhexyl) Phosphate Ionic Liquid as a Catalyst and Multifunctional Lubricant Additive: Synthesis, Optimization, Characterization, and Tribological Evaluation. Arab. J. Sci. Eng. 2024, 49 (6), 7995-8010.
[11]Wang, J.; Hu, W.; Li, J. Lubrication and Anti-Rust Properties of Jeffamine-Triazole Derivative as Water-Based Lubricant Additive. Coatings 2021, 11 (6), 679.
[12] Xiong, L.; He, Z.; Han, S.; Hu, J.; Xu, X.; Tang, J.; Wu, Y. Tribological Study of OH- and N-Containing Imidazoline Derivatives as Additives in Water-Glycol. Proc. Inst. Mech. Eng., Part J 2019, 233 (3), 466-480.
[13] Li S.; Pan Z.; Yang Y.; Li X.; Liu D. Research Progress of Benzotriazole and Its Derivatives in Lubricating Oil Corrosion Prevention. Lubricating Oil 2023, 38 (1), 23-27.
[14] Li Q.; Wang Y.; Yan F.; Guo Y. Characterization of Methyl-benzotriazole Amine Salt. Petroleum Processing and Petrochemicals 2016, 47 (7), 82-85.
[15] Wang, J.; Xu, X.M.; Zhou, J.L.; Zhao, Y.N.; Sun, X.L.; Tang, Y.; He, S.F.; Yang, H.M. Synthesis of New Sulfur-free and Phosphorus-free Ether-ester and Study on Its Properties As Ashless Friction Modifier. Acta Chim. Sinica 2023, 81, 461-468.
- The research gap is not properly identified and explained in the introduction section.
Response: Although benzotriazole derivatives display good tribological performance as lubricating additives, the poor solubility in hydrocarbons limits their applications. To improve the compatibility and functional performance of benzotriazole in hydrocarbons, a novel methylbenzotriazole-amide derivative (MeBz-2-C18) was innovatively designed and synthesized, using ethylamine as the linkage in this study. We have identified and explained the research gap in the revised introduction.
- The main objectives and innovation of the paper must be written in a clearer and more concise way at the end of the introduction section.
Response: Thanks very much for your suggestion. We have rewritten the main objectives and innovations of the research in a clearer and more concise way at the end of the introduction. "In this study, aiming to improve the compatibility and functional performance of benzotriazole in hydrocarbons, a novel methylbenzotriazole-amide derivative (MeBz-2-C18) was innovatively designed and synthesized, using ethylamine as the linkage. The structure of MeBz-2-C18 was characterized by nuclear magnetic resonance (NMR), high-resolution mass spectrometry (HR-MS), Fourier-transform infrared spectroscopy (FT-IR) and thermogravimetric analysis (TGA). Subsequently, the storage stability and tribological behavior of MeBz-2-C18 and the commercially available T406 with similar composition were comparative evaluated in the low-viscosity synthetic base oil. The worn surfaces were analyzed via scanning electron microscope-energy dispersive spectrometer (SEM-EDS) and X-ray photoelectron spectroscopy (XPS). These experimental analyses were integrated with density functional theory (DFT) calculations to elucidate the micro-lubrication mechanism of MeBz-2-C18." Please see the last paragraph in the introduction of the revised manuscript.
- The "Materials and Methods" section should be located prior to the "Results" section.
Response: Thanks for your proposal. However, according to the writing requirements in the template of International Journal of Molecular Sciences, the "Results" section and the "Materials and Methods" section need be placed in Section 2 and Section 3, respectively.
- The description of the synthesis of MeBz-2-Etn and MeBz-2-C18 is brief and lacks detail. It would be helpful to provide more information about the reaction conditions, such as temperature, reaction time, and yield.
Response: Thank you very much for your advice. We have provided the detail information for the synthesis of MeBz-2-Etn and MeBz-2-C18, please see the "Synthesis of MeBz-2-Etn and MeBz-2-C18" in Section 3.2.
- Figures 1 and 2 lack explanations of their contributions to the findings. A brief description of each figure would improve understanding.
Response: We combined the original Figure 1 and Figure 2 into the new Figure 1 in the revised manuscript, and its description has been changed to "The comparison between MeBz-2-Etn and MeBz-2-C18: (a) 1H NMR and (b) FT-IR." The new figure is used to contrastively characterize the structural differences between MeBz-2-Etn and MeBz-2-C18. In the 1H NMR spectrum of MeBz-2-C18, the hydrogen signals of oleoyl alkyl and -CH=CH- appeared at 2.13, 2.01, 1.55, 1.26, 0.88 ppm and 5.34 ppm, respectively (in Figure 1a). Meanwhile, the -CH₂ directly connected to -NH underwent a significant downfield shift (from 3.36 ppm to 3.93 ppm) due to the introduction of C=O. As can be observed from FT-IR spectra in Figure 1b, compared with MeBz-2-Etn, there is no N-H stretching vibrations of primary amine at 3372.9 and 3295.8 cm-1 in MeBz-2-C18. However, the C=O(NH) characteristic stretching vibration of secondary amide appears at 1643.5 cm-1, and the N(CO)-H and C(O)-N(H) stretching vibration peaks of secondary amide appear at 3309.7, 3071.7 and 1265.1 cm-1, respectively. Moreover, the peak located at 721.8 cm-1 can be attributed to the in-plane bending vibration of long alkyl chain -(CH2)n-. Collectively, these spectral characteristics serve as evidence for the successful grafting of oleoyl group in MeBz-2-C18.
- The manuscript mentions several figures and tables, such as Figure S3 and Table S3, that are only found in the supplementary file and not included in the main text. This discrepancy may confuse readers and should be addressed for clarity.
Response: We have modified the figures and tables mentioned in the manuscript to the main text, while the original spectra of the structural characterizations of MeBz-2-Etn and MeBz-2-C18 are still placed in the supplementary file.
- The supplementary file should be integrated into the manuscript for better coherence and accessibility.
Response: Thank you for your suggestion, we have integrated the supplementary file into the revised manuscript.
- The importance of additive-base oil compatibility for storage stability is not sufficiently explained.
Response: Thank you for your kindly reminder. We have explained the importance of additive-base oil compatibility for storage stability in the revised manuscript. "The storage stability of lubricants is a decisive factor influencing the compatibility between additives and base oils, which also serves as a fundamental prerequisite for practical applications. Therefore, the storage stability of oil samples at room temperature was investigated to study the compatibility of MeBz-2-C18 in the synthetic base oil, which was also compared with the commercially available T406 of similar composition."
- Figure 3 and Table S1 are referenced without explanation, and the implications of the compatibility results are not discussed.
Response: We have improved the explanation and discussion of the compatibility results in the revised manuscript. "As shown in Figure 3a, the oil samples with 0.25-1 wt.% additions of MeBz-2-C18 remained clear and transparent after being stored for 30 days, which is similar to the base oil (0 wt.%). In contrast, the oil sample with 0.5 wt.% T406 became cloudy on the 7th day (Figure 3b), indicating that the compatibility between MeBz-2-C18 and the synthetic base oil is better than that of T406. This may be due to the fact that the bonding modes of T406 and MeBz-2-C18 are ionic bond and covalent bond, respectively. The relatively weaker polarity of the covalent bond leads to better compatibility with non-polar base oils."
- Provide background information on the commercial additive T406 and its relevance to the study. Explain the importance of comparing MeBz-2-C18 with T406.
Response: The commercial additive T406 is a commercially available additive, which is widely used in the lubricant industry to improve the anti-wear and anti-oxidation performance. The structure of T406 is benzotriazole oleamide salt. Its structural composition is similar to that of MeBz-2-C18, so it is selected to conduct the performance comparative evaluation.
- Present a more explicit comparison between MeBz-2-C18 and T406, highlighting the differences in COF and WSD percentages.
Response: Thank you for your advice. In the revised manuscript, we have refined the comparative description of MeBz-2-C18 and T406. The differences in COF and WSD percentages are also presented in Figure 5b. "Compared with the base oil (Figure 5b), the ave. COF of the oil with 0.5 wt.% MeBz-2-C18 decreased by 3.2%, while that of the oil with 0.5 wt.% T406 increased by 4.5%. The ave. WSD of the oil with 0.5 wt.% addition of MeBz-2-C18 and T406 was decreased by 21.6% and 13.9%, respectively. The results indicate that MeBz-2-C18 exhibits better friction-reducing and anti-wear performance than that of T406."
- Offer a clear explanation of Figure 5, its content, and how it supports the claims made in the text.
Response: A clearer explanation of Figure 5 can be found in Section 2.3.2.
"The tribological performance of MeBz-2-C18 was compared with T406, using 0.5 wt.% as the optimal addition. As shown in Figure 5a, the COF of the base oil suddenly increased and significant fluctuations occurred after 1200 s during the friction. However, the COF of the oils with 0.5 wt.% T406 or MeBz-2-C18 showed smaller variations, indicating that the addition of T406 and MeBz-2-C18 could stabilize the friction. Compared with the base oil (Figure 5b), the ave. COF of the oil with 0.5 wt.% MeBz-2-C18 decreased by 3.2%, while that of the oil with 0.5 wt.% T406 increased by 4.5%. The ave. WSD of the oil with 0.5 wt.% addition of MeBz-2-C18 and T406 was decreased by 21.6% and 13.9%, respectively. The results indicate that MeBz-2-C18 exhibits better friction-reducing and anti-wear performance than that of T406."
- Clearly explain the chemical changes observed in XPS analysis, including the increased C=O content and the appearance of N-O and C-N bonds in MeBz-2-C18 samples.
Response: We have explain the chemical changes observed in XPS analysis. "In the C1s spectra, there are four main peaks for MeBz-2-C18_wear, namely metal carbides (283.4 eV, ~2.9%), C-C/C=C (284.8 eV, ~72.9%), C-O/C-N (286.0 eV, ~20.4%) and C=O (288.5 eV, ~3.6%)[20-21]. In contrast, the Base oil-wear spectrum only exhibits C-C/C=C (284.8 eV, ~86.9%) and C-O/C-N (286.0 eV, ~13.1%). In the O1s spectra, peaks at 529.6 eV, 531.7/531.9 eV and 533.1 eV correspond to Fe-O, C=O and C-O bonds, respectively. Notably, the Fe-O and C-O content in MeBz-2-C18_wear increases compared with Base oil-wear, i.e. 30.0% vs. 13.0% and 6.7% vs. 0%, respectively. While the C=O content in MeBz-2-C18_wear decreases compared with Base oil-wear (63.3% vs. 86.9%). The results indicate that the amide bond were broken during friction, which further led to the formation of iron oxides. The peaks at 709.6 eV, 723.5 eV, 713.5 eV and 711.0 eV in the Fe2p spectra correspond to Fe (2p3/2), Fe²⁺ (2p1/2), Fe²⁺ (2p3/2) and Fe³⁺ (2p3/2), respectively. This suggests that local high temperatures and high loads caused chemical reactions between iron in steel balls and oxygen in air during friction[22-23]. Combined with the O1s spectra, it is evident that an iron oxide film was formed during friction, which might be composed of Fe₂O₃, FeOOH, FeO and Fe₃O₄[24-25]. Additionally, the N1s peaks for MeBz-2-C18_wear appear at 399.4 eV and 401.7 eV, corresponding to N-O and C-N bonds, which indicates that some amides were converted into nitrogen oxides[25-26]. The results demonstrate that the friction film formed by MeBz-2-C18 consists of organic oxides and iron oxides, which can enhance the anti-wear and friction-reducing properties."
- The comparison between MeBz-2-C18 and T406 is mentioned, but the significance of the differences in ESP values is not thoroughly discussed.
Response: Thank you for your question. We have discussed the significance of the differences in ESP values differences in ESP values for MeBz-2-C18 and T406. "As shown in Figure 7, the minimum and maximum ESP values for T406 and MeBz-2-C18 are -0.0475, 0.0565 and -0.0665, 0.0731 respectively. According to [27], the smaller the minimum value of ESP, the stronger adsorption capacity on the metal surface. This indicates that MeBz-2-C18 has a stronger adsorption ability on the metal surface compared with T406."
- Section 3.2: The text is presented as a continuous block of text without clear headings, subheadings, or bullet points, making it difficult to read and comprehend. Improving the formatting and structure would make the text more accessible and user-friendly.
Response: Thank you very much for your suggestion. I have modified the presentation and added two subheadings in the revised Section 3.2, to make it easier for readers to read and understand.
- The conclusions section requires significant improvements and should be effectively rewritten after addressing the aforementioned comments.
Response: Thanks for your advice. I have rewritten the conclusions section, addressing the aforementioned comments.
- The paper shall be logically rearranged and rephrased to help understanding the main point of this work, as well as highlighting novelty. The current manuscript is more like a technical report rather than a research paper. The research contents are listed linearly, without highlighting the focus and innovation.
Response: Thank you very much for your careful review and valuable suggestions. We have have rearranged and rewritten the content in a logical way to highlight the novelty and main point.

Reviewer 2 Report
Comments and Suggestions for Authors
Since all the review has been received already. Please let me know if you need my comments.
Author Response
No comments.
Response : Thank you very much for your review and affirmation of this work.
Reviewer 3 Report
Comments and Suggestions for Authors
The list of comments is attached.

The sub-headings were not written grammatically. Kindly rewrite the sub-headings.
Author Response
- Figure 4 and figure 5 look the same. Kindly correct it.
Response: Thank you very much for your careful review. We have corrected Figure 4 in the revised manuscript.
- The abbreviation MeBz-2-Etn was not mentioned in its full form. Kindly, write the abbreviation along its full form when discussed very first.
Response: Thanks for your kind reminder. We have added the full form of MeBz-2-Etn when it is first mentioned.
- 2.3.1 Different additions of MeBz-2-C18, the sub-heading is not clear. Kindly rewrite the heading.
Response: According to your proposal, We have modified the sub-heading of 2.3.1 to "Different MeBz-2-C18 additions".
- The abbreviation TLC was not written in its full form. Kindly write the full form the abbreviation.
Response: Thank you for your kind reminder. We have added the full form of TLC when it is first mentioned.
- In para 244, the breakdown of amide bonds was discussed. Kindly support the argument with reference.
Response: In our previous research [1-2], we found that polar amide and ester bonds are more likely to break during friction. This reduces the interfacial shear force and enables easy reactions with metal surfaces to form iron oxide films, thus showing better anti-wear and friction-reducing performance.
- Wang, J.; Xu, X.M.; Zhou, J.L.; Zhao, Y.N.; Sun, X.L.; Tang, Y.; He, S.F.; Yang, H.M. Synthesis of New Sulfur-free and Phosphorus-free Ether-ester and Study on Its Properties As Ashless Friction Modifier. Acta Chim. Sinica 2023, 81, 461-468.
- Xu, X.; Yang, F.; Yang, H.; Zhao, Y.; Sun, X.; Tang, Y. Preparation and tribological behavior of Sulfur- and Phosphorus-Free Organic Friction Modifier of Amide–Ester Type. Lubricants 2024, 12 (6), 196.
- In figure 4(b), the ave. WSD graph as three images. However, the image scale was not mentioned. Kindly put image scale in the three pictures.
Response: Thank you for your suggestion. We have put image scale to the wear scar images in Figure 4b.
- Heading 2.2.2 Storage stability in Base oil is not discussed in depth. Does the change in transparency result in the degradation of the additive? Kindly, discuss the change in transparency of the additive.
Response: Thank you for your question. We have reorganized the discussion in Section 2.2.2. Combined with the results of 2.2.1 Thermal Stability, the initial decomposition temperatures of T406 and MeBz-2-C18 are 76.8 °C and 185.5 °C respectively, both of which are higher than the storage temperature of the oil sample (room temperature, ~30 °C). This indicates that during storage at room temperature, neither T406 nor MeBz-2-C18 will decompose. Their transparency is not related to degradation, but only to solubility.

Round 2
Reviewer 1 Report
Comments and Suggestions for Authors
This revision has significantly improved the manuscript. The authors have thoroughly addressed all of my previous comments and concerns, and I am satisfied with the revisions they have made to the manuscript.